# Interleukin (IL)-22 from IL-20 Subfamily of Cytokines Induces Colonic Epithelial Cell Proliferation Predominantly through ERK1/2 Pathway

**DOI:** 10.3390/ijms20143468

**Published:** 2019-07-15

**Authors:** Md. Moniruzzaman, Ran Wang, Varinder Jeet, Michael A. McGuckin, Sumaira Z. Hasnain

**Affiliations:** 1Faculty of Medicine, The University of Queensland, Brisbane, QLD 4067, Australia; 2Immunopathology Group, Mater Research Institute–The University of Queensland, Translational Research Institute, Brisbane, QLD 4102, Australia; 3Australian Prostate Cancer Research Centre–Queensland, Institute of Health Biomedical Innovation (IHBI), Queensland University of Technology, Brisbane, QLD 4102, Australia; 4Macquarie University Centre for the Health Economy, Macquarie University, NSW 2109, Australia; 5Faculty of Medicine, Dentistry and Health Sciences, the University of Melbourne, VIC 3052, Australia; 6Australian Infectious Disease Research Centre, University of Queensland, Brisbane, QLD 4067, Australia

**Keywords:** cell proliferation, wound healing, IL-20, IL-22, IL-24, ERK1/2

## Abstract

The interleukin (IL)-20 subfamily of cytokines consists of IL-19, IL-20, IL-22, IL-24, and IL-26, and the expression of IL-20, IL-22, and IL-24 is reported to be higher in the colon of patients with ulcerative colitis. Although the receptors for these cytokines are highly expressed in the colon epithelium, their effects on epithelial renewal are not clearly understood. This study evaluated the effects of IL-20, IL-22, and IL-24 in epithelial renewal using the LS174T human colon cancer epithelial cell line. LS174T cells were treated with IL-20, IL-22, and IL-24 (25, 50, and 100 ng/mL) and a live-cell imaging system was used to evaluate the effects on cell proliferation. Following treatment, the signaling pathways contributing to cell proliferation were investigated through Western blotting in LS174T cells and downstream transcriptional changes through qRT-PCR in LS174T cells, and RNA-Seq in primary murine intestinal epithelial cells. Our results demonstrated that only IL-22 promoted LS174T cell proliferation, mediated via extracellular-signal-regulated kinase (ERK)1/2-mediated downstream regulation of p90RSK, c-Jun, and transcriptional changes of *TRIM15* and *STOM*. IL-22 also promoted expression of ERK1/2-independent genes such as *DDR2*, *LCN2*, and *LRG1*, which are known to be involved in cell proliferation and migration. This study suggests that IL-22 induces cell proliferation in highly proliferative cells such as intestinal epithelial cells.

## 1. Introduction

The gastrointestinal tract is the largest organ that is continuously exposed to dietary antigens and commensal microbiota. The specialized gut mucosal epithelial cells therefore form and secrete a physical and biochemical barrier that segregates the host and microorganisms. The intestinal epithelial cells also sense and respond to microbial stimuli and coordinate appropriate immune responses, playing a fundamental role in developing and maintaining mucosal immunoregulatory function [1,2,3]. Thus, preserving and renewing the intestinal epithelial barrier is necessary to conserve immune homeostasis in the gut.

The interleukin (IL)-20 subfamily of cytokines belongs to the large IL-10 family and consists of IL-19, IL-20, IL-22, IL-24, and IL-26. These cytokines are primarily produced by leukocytes but preferentially act on non-hematopoietic cells, particularly epithelial cells [4]. IL-20, IL-22, and IL-24 have been found to be higher in the colons of patients with ulcerative colitis, where mucosal epithelial damage is very common [4,5]. In addition, there is data demonstrating that IL-20, IL-22, and IL-24 can act on epithelial cells and modulate their function. These cytokines act through the heterodimeric receptor complexes of IL-20RA/IL-20RB and IL-22RA/IL-20RB (IL-20 and IL-24), and IL-22RA/IL-10RB (IL-22). Although the cytokines can bind with an individual subunit, binding with the heterodimeric complex is necessary for appropriate signal transduction [4]. These receptor subunits, especially IL-22RA, are highly expressed in the colonic epithelium but not expressed on immune cells. IL-20RA and IL-20RB are also expressed in the colonic epithelium; however, IL-20RB can also be found on immune cells whereas IL-20RA is mostly absent on immune cells [6]. A few studies have reported that IL-22 can activate ileal epithelial stem cells to promote epithelial regeneration in the small intestine [7,8]. In contrast, Zwarycz et al. have demonstrated an inhibitory action of IL-22 in ileal epithelial stem cell expansion using the organoid model [9]. A recent report showed that IL-22 is a master regulator of DNA damage response machinery in the colon that protects colonic stem cells from genotoxic stress [10]. However, the role of IL-20 and IL-24 in colonic health, especially in epithelial renewal and wound healing, is not clearly understood.

Therefore, this study aimed to mechanistically investigate the potential role of IL-20, IL-22, and IL-24 in colonic epithelial renewal, and in particular, the effects on cell proliferation. We treated the colonic epithelial and goblet cell like cell line, LS174T, with IL-20, IL-22 or IL-24 and assessed their effects on cell proliferation. We also investigated the signaling pathways activated by these cytokines and their contribution to cell proliferation. The results demonstrated that only IL-22 could induce LS174T cell proliferation and that proliferation was primarily controlled by activation of the ERK1/2 signaling pathway.

## 2. Results

### 2.1. IL-22 but not IL-20 and IL-24 Promote Proliferation of Colonic Epithelial LS174T Cells

We evaluated whether IL-20, IL-22, and IL-24 can induce proliferation of LS174T cells. Using a live-cell-imaging system we found that at the concentrations tested only IL-22 significantly promoted LS174T cell proliferation (Figure 1B) compared to the unstimulated controls. The greatest effect was observed with 50 and 100 ng/mL of IL-22 starting within 4 h of treatment, where 25 ng/mL concentrations had moderate effects starting from 12 h post treatment (Figure 1B), demonstrating a dose-dependent effect. In contrast, treatment with IL-20 or IL-24 did not affect LS174T cell proliferation (Figure 1A,C). We confirmed that the respective receptor subunits of IL-20 and IL-24, IL-20RA and IL-20RB, were expressed in the LS174T cells (data not shown).

### 2.2. IL-22 Activates JAK-STAT, Akt, and Mitogen-activated protein kinase (MAPK) Pathways in Intestinal Epithelial Cells

Next, we investigated the earliest signalling pathways activated by IL-22 in LS174T cells. As depicted in Figure 2, IL-22 at 50 ng/mL activated STAT1 (Figure 2A), STAT3 (Figure 2B), STAT5 (Figure 2C), Akt (Figure 2D), ERK1/2 (Figure 2E) and p38 MAPK (Figure 2F) in LS174T cells at 15 and 30 min of treatment. However, NF-κB (nuclear factor kappa-light-chain-enhancer of activated B cells) pathways remained unresponsive to IL-22 treatment (Appendix A). To confirm these changes in non-malignant cells we treated murine intestinal epithelial cells (mIECs) with IL-22 (100 ng/mL). IL-22 similarly activated STAT3, Akt, and ERK1/2 pathways in the mIECs (Appendix A), whereas STAT1, STAT5, p38, and NFκBp65 were irresponsive to IL-22 (Appendix A). We observed a trend of changes in the downstream of the ERK1/2 pathways p90RSK and c-Jun (Appendix A). To confirm the activation of these pathways, RNA-Seq was conducted on mIECs treated with IL-22. The kyoto encyclopedia of genes and genomes (KEGG) pathway and gene ontology analyses of the RNA-Seq data revealed the positive regulation of genes associated with JAK-STAT, Akt, and ERK1/2 pathways in the mIECs (Appendix A).

### 2.3. IL-22-Mediated Cell Proliferation Is Predominantly Controlled via the ERK1/2 Signaling Pathway

To assess the involvement of STAT1 (Figure 3A), STAT3 (Figure 3B), STAT5 (Figure 3C), Akt (Figure 3D), ERK1/2 (Figure 3E), and p38 MAPK (Figure 3F) pathways in IL-22-induced cell proliferation, we pre-treated LS174T cells with specific inhibitors for each pathway for 1 h. Of note is that none of the inhibitors alone affected cell proliferation (Figure 3). Cells were then stimulated with 50 ng/mL of IL-22 and cell proliferation was assessed in the presence or absence of the inhibitors. Although STAT3 is well-known to induce epithelial regeneration in the small intestine [8], we did not observe any influence of STAT3 signalling on LS174T cell proliferation (Figure 3B). Interestingly, in the presence of ERK1/2-specific inhibitor (Figure 3E) IL-22-mediated effects of cell proliferation were markedly reduced. None of the other inhibitors tested altered IL-22-induced cell proliferation, indicating that LS174T cell proliferation is substantially dependent on ERK1/2.

### 2.4. p90RSK and c-Jun Are the Downstream Regulators of IL-22-Mediated ERK1/2 Signaling Pathway

The ERK1/2 pathway had an impact on IL-22-mediated LS174T cell proliferation and we therefore sought to understand the downstream regulatory signaling pathways activated by ERK1/2. First, to validate the action of ERK1/2 inhibitor, we measured the phosphorylation of ERK1/2 and found a marked decrease in ERK1/2 phosphorylation compared to control and IL-22 treatment (Figure 4A). Treatment with IL-22 (50 ng/mL) alone significantly increased the phosphorylation of 90 ribosomal s6 kinase (90RSK) (Figure 4B) and c-Jun (Figure 4C). However, pre-treatment with ERK1/2 inhibitor blocked IL-22-mediated activation of both p90RSK and c-Jun, confirming these molecules as the downstream effectors of ERK1/2. Activation of ERK1/2 and its downstream signalling pathways are required for cellular survival, explaining the changes in the basal levels of p90RSK and c-Jun with ERK1/2 inhibition (Figure 4A,B).

### 2.5. IL-22-Modulated Pathways Are Associated with Epithelial Cell Proliferation

IL-22 has been shown to modulate ER stress and oxidative stress in secretory cells [11]. Moreover, IL-22 has been shown to induce goblet cell hyperplasia in nematode infection [12]. In addition to the changes observed in LS174T cell proliferation with IL-22, we aimed to investigate the effects of IL-22 on secretory cell differentiation pathways, and ER and oxidative stress pathways. Interestingly, IL-22 treatment decreased the expression of IL-22RA1 at 12 h (Appendix A). In contrast to IL-22-mediated effects in pancreatic beta cells [11], 12 and 24 h IL-22 treatment increased the gene expression of ER and oxidative stress markers GRP78 (Figure 5A), sXBP1 (Figure 5B) and NOS2 (Figure 5C). The induction of ER stress markers with IL-22 treatment was not detrimental to the viability of the cells (Appendix A). IL-22 also decreased the expression of the intestinal mucin, MUC2 (Figure 5D) and the protein disulphide isomerase AGR2 (Figure 5E), indicating a reduced secretory function of these cells. However, no major changes were observed in the goblet cell differentiation factors ATOH1 (Figure 5G) and SPDEF (Appendix A). HES1, a marker for enterocytes, decreased with IL-22 treatment (Figure 5F), though, interestingly, the ratio of ATOH1 and its antagonist HES1 increased at 12 h with IL-22 treatment (Figure 5H), indicating that IL-22-induced proliferation is directed towards an increase in the enterocytes rather than goblet cells.

### 2.6. IL-22 Increases the Expression of Cell Proliferation and Migration-Associated Genes in Primary Mouse Intestinal Epithelial Cells

To identify the contributing transcriptional changes in IL-22-mediated increases in cell proliferation, we used an unbiased approach and conducted RNA-sequencing in primary mIECs. Phosphorylation of STAT3 validated the action of IL-22 in mIECs at 4 h of treatment (Figure 6A). Gene ontology analyses (list of genes sorted from false discovery rate ≤ 0.05 and fold change ≥ 2) showed several genes were differentially upregulated with IL-22 treatment which are associated with cellular proliferation and migration. Importantly, we found the glucose transporter *Stom* was differentially expressed with IL-22 treatment (Figure 6B). Validation using qRT-PCR revealed that *Ddr2* (Figure 6C), *Lrg1* (Figure 6D), *Timp1* (Figure 6E), *Trim15* (Figure 6F), *Reg3β* (Figure 6G), *Lcn2* (Figure 6I), and *Stom* (Figure 6J) expression was induced by IL-22. These results confirm that IL-22 promoted cell proliferation in the intestine which is mediated through modulation of the aforementioned genes. In addition, we also observed an upregulation of the genes associated with cell differentiation with IL-22 treatment (Appendix A). This can be correlated with IL-22 function in promoting gene expression of antimicrobial peptides from regenerating (REG) family member such as *Reg3α* (Figure 7C), *Reg3β* (Figure 6G), and *Reg3γ* (Figure 7D).

### 2.7. IL-22-Induced Effects Are Mediated through ERK1/2-Dependent and -Independent Transcriptional Modifications

Inhibition of the ERK1/2 signaling pathway partially decreased IL-22-induced GRP78 expression (Figure 7A). ERK1/2 inhibitor alone significantly increased expression of *MUC2* (Figure 7B), *HES1* (Figure 7E), and *ATOH1* (Figure 7F), which is suggestive of a more dominant secretory cell phenotype when the ERK1/2 pathway was inhibited. Although ERK1/2 inhibition had no effect on basal expression of the secretory antimicrobial peptides *REG3α* and *REG3γ*, IL-22-driven expression of these genes was boosted by ERK1/2 inhibition in the LS174T cells (Figure 7C,D) and primary mIECs (Appendix A). We did not observe any changes in the expression of *sXBP1*, *NOS2*, and *AGR2* with ERK1/2 inhibitor (Appendix A). We found that IL-22 promoted the expression of cell proliferation and migration-associated genes including *DDR2* (Figure 7H), *LRG1* (Figure 7I), *TRIM15* (Figure 7J), and *LCN2* (Figure 7K) and glucose transporter *STOM* (Figure 7L). Inhibition of the ERK1/2 pathway only reversed the expression of *TRIM15* (Figure 7J) and *STOM* (Figure 7L) in LS174T cells. However, IL-22-mediated increases in *Trim15*, *Stom*, and *Lrg1* were reduced by ERK1/2 inhibition in primary mIECs (Appendix A) suggesting that ERK1/2 at least in part is responsible for controlling IL-22-mediated changes in proliferation in intestinal epithelial cells. No changes were observed with ERK1/2 signalling inhibition in the expression of *Lcn2* in the mIECs (Appendix A) and matrix metalloprotease 7 (*MMP7*) and valosin containing protein (*VCP*), which are known to be involved in cell cycle and proliferation (Appendix A). The anti-inflammatory gene *SOCS3* also did not change with ERK1/2 inhibition (Appendix A).

## 3. Discussion

Mucosal epithelial renewal can be achieved by promoting cell proliferation, migration, and differentiation, which are controlled by the temporal and spatial activation of multiple signaling pathways [13]. Activation or inhibition of signaling pathways modulate the function of a number of transcription factors; however, the cumulative effects of a myriad of signaling crosstalk and transcriptional modulation determine cell fate [14,15]. Intestinal epithelial cells are fast-proliferating cells and complete turnover occurs in 3.48 ± 1.55 days [16]. This rapid and dynamic turnover of the epithelium is critically regulated by multiple factors in the environment of the stem cell/transit-amplifying cell niche including hormones, cytokines, and chemokines secreted from the immune cells and peripheral organs [17]. The IL-20 subfamily of cytokines has attracted interest in the last decade because of their unique biology in the epithelium. Their receptors are highly expressed in the epithelial cells but are absent or less expressed in the parent immune cells [4,6]. It has been found that the expression levels of IL-20, IL-22, and IL-24 are higher in the colon of patients with ulcerative colitis [4] but that their roles in colonic epithelial renewal are not clearly understood. Lindemans et al. have reported that gene delivery of IL-22 promoted stem-cell-mediated small intestinal epithelial regeneration [8] while others showed the transfer of IL-22 producing immune cells to improve dextran sodium sulphate-induced colitis [18,19]. Our study has demonstrated that among the studied IL-20 subfamily of cytokines, IL-22 is the only cytokine effective at promoting proliferation of LS174T colonic epithelial cells, whereas IL-20 and IL-24 exerted no effects, despite the receptor subunits being expressed. Moreover, although IL-20 and IL-24 activated downstream signaling in LS174T cells (data not shown), we did not see any effects on cell proliferation. We speculate that IL-20 and IL-24 affect different cellular pathways in the intestinal epithelial cells, differentiating them from the functions of IL-22. These results have some similarities with the report by Koluman et al., who showed that although IL-20, IL-22, and IL-24 are all highly expressed in wounded skin, only IL-22 has a major effect on wound repair and cell proliferation [20]. Other studies also have demonstrated that IL-22 is the most potent cytokine in this family in promoting hyperproliferation, cell differentiation, antimicrobial responses, and positive regulation of genes to promote tissue remodeling in psoriasis [21,22].

When investigating the downstream intracellular mechanism(s) of action, previous studies have shown IL-22 to differentially activate JAK-STAT, Akt, and MAP kinase pathways in various cell types [5,23], but that the transcription factor STAT3 is the key regulator of IL-22-mediated action [23]. To this end, a recent report has highlighted that IL-22-mediated phosphorylation of STAT3 plays a critical role in proliferation of Lgr5+ stem cells in the small intestine, thus promoting ileal epithelial regeneration [8]. In contrast, our study in human cells demonstrated that not STAT3 but only ERK1/2 signaling drove LS174T colonic epithelial cell proliferation (Figure 8). Our finding is in contrast to other reports which demonstrate the important role of STAT3 in IL-22-mediated mucosal wound healing in mouse colon epithelial cells [7]. In addition, activation of Akt has been reported to inhibit caspase-3 mediated apoptosis in the intestinal epithelial cells [24], where Akt along with ERK1/2 and p38 have been shown to be differentially involved in fibroblast, hepatocyte, and airway smooth muscle cell proliferation and migration [25,26,27,28]. However, we did not see any influence of Akt in IL-22-mediated proliferation. Our results suggest that rather than STAT3, the ERK1/2 pathway could be one of the leading mechanisms underlying IL-22-driven proliferation. Herein, we found that the p90RSK, a serine/threonine kinase, is upregulated by IL-22 and controlled by ERK1/2. p90RSK is known to be a downstream effector of MAP kinases [29] and has been shown to play a critical role in cell survival by regulating cell cycle checkpoints, thus promoting cell proliferation [30,31,32]. In addition, we found that ERK1/2 also governs the activation of c-Jun, which is a component of activator protein (AP)-1 [33,34]. AP-1 is a dimeric transcription factor known to be involved in cell proliferation, differentiation, survival, and apoptosis. Being a component of AP-1, c-Jun has been reported to promote cell proliferation through inhibition of p16 and p21 and induction of cyclin D1 [35]. Therefore, it is conceivable that p90RSK and c-Jun are the downstream transcriptional mediators responsible for ERK1/2-dependent IL-22-mediated LS174T cell proliferation.

Focal adhesions are the multi-protein complexes which are known to promote cell development, migration, and wound healing [36,37,38]. We found that treatment with IL-22 increased the expression of *TRIM15*, which encodes a tripartite motif (TRIM) protein family member that contributes to focal adhesion formation. It has been shown that siRNA knockdown of *TRIM15* causes impaired migration and reduced disassembly of focal adhesions in human epithelial cells. This hampers the dynamic and rapid turnover of the focal adhesions that can potentially affect cell growth and epithelial wound healing [37]. In support, other reports have shown that higher expression of TRIM15 is required for the survival of the mucosal epithelial tissues, particularly in the stomach and colon, where decreased expression promotes tumour growth in these organs [38,39]. Therefore, in keeping with the findings of this literature, it is inferred that IL-22-induced *TRIM15* expression could be one of the contributing factors in the observed LS174T cell proliferation. In addition, we found that IL-22 promoted the expression of the stomatin gene *STOM*, which is well-known for its modulatory actions of acid-sensitive ion channels [40]. However, it has also been shown to interact with glucose transporter 1 (GLUT1) and inhibit its functionality in the plasma membrane raft [41]. In contrast, Kumar et al. have shown that the binding of stomatin with GLUT1 does not alter the transporter activity, but rather serves as an anchor to favor GLUT1 activation [42]. It is well-known that the synthesis of cellular components including nucleic acid, lipids, and proteins to promote cell proliferation depends on the availability of nutrients such as glucose. Therefore, it is possible that IL-22 also induces GLUT1 activity through stomatin to promote LS174T cell proliferation. Our results also demonstrated that inhibition of IL-22-induced ERK1/2 signaling significantly reversed the expression of *TRIM15* and *STOM* in LS174T cells and primary mIECs. Therefore, it is possible that IL-22-induced cell proliferation is mediated through ERK1/2 controlled expression of *TRIM15* and *STOM,* which could impact on wound healing.

Several other genes which are independent of the ERK1/2 pathway but are directly related to cell proliferation, migration, and wound healing were also upregulated with IL-22 treatment (for example, *DDR2*, which is a receptor tyrosine kinase reported to be activated by collagen I, II, III, V, and X [43]). Once activated, DDR2 was found to trigger the ERK1/2 signaling pathway and promote matrix metalloprotease 2 and 9 expression to contribute to cell proliferation, migration, and wound healing [44,45,46]. As the ERK1/2 signaling pathway also can be controlled by *DDR2*, which explains why we did not observe any changes in *DDR2* expression with the inhibition of the ERK1/2 signaling pathway. Another gene that was upregulated with IL-22 treatment is *LRG1*, a gene for leucine rich alpha2-glycoprotein 1. *LRG1* was reported to promote endothelial cell proliferation [47] and differentiation of granulocytes [48]. Therefore, it is possible that along with the endothelial cells and chondrocytes, these genes also contributed in the LS174T epithelial cell proliferation and differentiation.

The upregulation of ER stress markers *GRP78* and *sXBP1* with IL-22 was in contrast to the effect of IL-22 reported in other secretory cells [11,49]. However, it highlighted the role of IL-22 in inducing cell proliferation, which may also induce cellular stress in highly proliferative cells [50]. This role of IL-22 in inducing proliferation may be detrimental during malignant transformation and tumor progression, as reported previously [51]. Here, IL-22 was seen to increase the expression of *LCN2* that encodes the protein lipocalin 2. Lipocalin-2 is a microbiota-inducible innate immune protein which regulates intestinal homeostasis. LCN-2 knockout mice harbor colitogenic bacteria, and this protein has been reported to reduce intestinal inflammation in the IL-10-knockout model of colitis [52,53]. It has been found that ER stress upregulates the expression of *LCN2* [54], where this gene has been reported to contribute to mitochondrial biogenesis, proliferation, and autophagy in vitro [55]. Therefore, it is possible that IL-22-regulated ER stress and subsequent upregulation of *LCN2* also contribute to the observed proliferation. Moreover, due to increased IL-22-induced proliferation we observed a decrease in MUC2 and goblet cells-associated factors *AGR2* and *HES1*. Interestingly, inhibition of ERK1/2 decreased expression of *GRP78* and increased the *ATOH1*/*HES1* ratio without affecting *MUC2* expression. These findings indicate that IL-22 affects the secretion of secretory proteins other than MUC2, particularly the regenerating islet-derived 3 family antimicrobial peptides *REG3α* and *REG3γ*. Although these peptides help in maintaining barrier integrity, overexpression of *REG3α* has been found to promote insulinoma cell growth through increases in cyclin D1 and CDK4 levels [56], where *REG3γ* promotes β cell regeneration in type I diabetic mice [57]. In injured skin, REG3α has also been found to induce keratinocyte proliferation and differentiation to promote wound healing [58]. Therefore, it is possible that IL-22-induced production of REG family members also partially contributes to the proliferation of LS174T cells and may be controlled by other activation pathways such as STAT3.

Taken together, this study has revealed that IL-22 is the only IL-20 family cytokine able to induce colonic epithelial cell proliferation. IL-22-driven proliferation is controlled by ERK1/2-mediated activation of p90RSK and c-Jun and both ERK1/2-dependent and-independent activation of downstream effectors which directly contribute to cell proliferation, migration, and wound healing (Figure 8). These results suggest that IL-22 could be a potential target to promote cell proliferation and wound healing in the damaged colonic epithelium associated with different acute and chronic intestinal diseases. However, further studies are required to justify its clinical applicability exploring the cytokine’s effectiveness, mechanism(s) of action, and possible adverse effects in vivo.

## 4. Materials and Methods

### 4.1. Chemicals and Reagents

Dulbecco’s modified Eagle medium (DMEM) and Dulbecco’s phosphate buffered saline (DPBS), fatal bovine serum (FBS), TrypLE^TM^ Express, pen-strep, glutamax, and protease inhibitor were purchased from Gibco Life Technologies (Grand Island, NY, USA). Human (h) IL-22, hIL-24, and all phosphorylated and secondary antibodies (Section 4.2) were procured from Cell Signaling Technology (Danvers, MA, USA). β-actin was purchased from Novus Biologicals (Littleton, CO, USA). The phosphatase inhibitor PhosSTOP was obtained from Roche (Basel, Switzerland). hIL-20 was purchased from R&D Systems (Minneapolis, MN, USA) and mIL-22 from PeproTech (Rocky Hill, NJ, USA). The specific inhibitor of STAT1 (Fludarabine) was purchased from Santa Cruz Biotechnology, Inc. (Dallas, TX, USA), those of STAT3 and STAT5 from Calbiochem (San Diego, CA, USA), PI3K/Akt inhibitor (LY294002) and MEK1/2-ERK1/2 inhibitor (U0126) from Sigma-Aldrich (St. Louis, MO, USA), and p38 MAP kinase inhibitor (SB203580) from Tocris Bioscience (Bristol, UK).

### 4.2. Antibodies Used in This Study

The antibodies used in this study were p-STAT1 (Tyr701; 58D6; 1:1000), p-STAT3 (Tyr705; D3A7; 1:2000), p-STAT5 (D47E7; 1:1000), p-ERK1/2 (Thr202/Tyr204; E10; 1:1000), p-p38 (Thr180/Tyr182; 12F8; 1:1000), p-Akt (Ser473; 1:1000), p-NF-κBp65 (Ser536; 93H1; 1:1000), p-c-Jun (Ser73; 1:1000), p-p90RSK (Ser380; 1:1000), β-actin (AC-15; 1:5000), anti-rabbit IgG (H+L; 1:30000), and anti-mouse IgG (H+L; 1:15000).

### 4.3. Cell Culture

Human colon adenocarcinoma cells (LS174T from the European Collection of Cell Culture, Salisbury, UK) were cultured in DMEM supplemented with 10% heat-inactivated FBS, 1% glutamate, and 1% pen-strep. Cells were washed with PBS, treated with TrypLE^TM^ Express for 3–5 min, and subsequently trypsin was neutralized using culture medium. Collected cells were then centrifuged at 800 rpm for 3 min to remove the trypsin, resuspended in the culture medium, and plated according to the necessities of the experiments.

### 4.4. Live-Cell Imaging for Cell Proliferation

LS174T cells (1.0 × 10^4^ cells/well) were plated in 96-well plates (Corning) and incubated overnight at 37 °C, allowing them to settle down. Cells were then treated with either PBS or 25, 50, or 100 ng/mL concentrations of IL-20, IL-22, or IL-24 in a serum-free medium, and immediately placed in the IncuCyte^®^ Live Cell Analysis System (Essen Bioscience, Ann Arbor, MI, USA). Cell proliferation was then observed and automatically analyzed from captured pictures taken every two hours for a consecutive 36 h.

To identify the impacts of IL-22-activated different signaling pathways, cells were pre-treated with 50 μM of STAT1 inhibitor (fludarabine), 50 μM of STAT3 inhibitor VI, 50 μM of STAT5 inhibitor, 10 μM of PI3K/Akt inhibitor (LY294002), 10 μM of MEK1/2 inhibitor (U0126), or 10 μM of p38 MAP kinase inhibitor (SB203580) for 1 h. Then, the cells were stimulated with IL-22 (50 ng/mL) and observed in the IncuCyte^®^ Live Cell Analysis System for 36 h.

### 4.5. Isolation and Maintenance of Primary Murine Intestinal Epithelial Cells

mIECs were isolated and cultured from three wild type C57BL/6 mice following the protocol established by Miyoshi et al. [59]. Briefly, the mouse colon was removed, longitudinally opened, and washed with ice-cold PBS containing 1% pen-strep to remove all fecal material. Tissue of approximately a centimeter in size was then cut and treated with ice-cold 8 mM Ethylenediaminetetraacetic acid (EDTA)/PBS containing 1% pen-strep for 1 h at 4 °C. Subsequently, tissue was incubated with 2 mg/mL collagenase and 50 μg/mL gentamycin in F12-DMEM containing 10% FBS, 1% glutamax, 1% HEPES (4-(2-hydroxyethyl)-1-piperazineethanesulfonic acid), and 1% pen-strep (washing medium) for 5–10 min at 37 °C. Crypts were then isolated using a 10 mL medium through 30 s vortexing and repeating the process five times. Isolated crypts were collected through washing and centrifugation at 500 rpm for 5 min at 4 °C and plated in 24-well plate following Matrigel embedding.

### 4.6. Western Blotting

LS174T (2 × 10^6^ cells/well) cells were plated in a 6-well plate and reached at least 70% confluency. The cells were then treated with either PBS (control) or IL-22 (50 ng/mL) for 15 or 30 min. As the mIECs were cultured in Matrigel, we threated mIECs with 100 ng/mL of IL-22 for 1 h. Then the cells were lysed with RIPA (Radioimmunoprecipitation assay) buffer (50 mM Tris-HCI, pH 7.5, 150 mM NaCI, 1.0% Nonidet P-40, 0.1% sodium deoxycholate) supplemented with phosphatase and protease inhibitors (Roche, Basel, Switzerland). Protein concentrations in the lysates were determined using a BCA (bicinchoninic acid) assay kit (Thermo Scientific) and samples were stored at −80 °C until use. Twenty μg of proteins from each sample were resolved using NuPAGE 4–12% Bis-Tris protein gels (Invitrogen^TM^; Seventeen Mile Rocks, QLD, Australia) at 100 V and transferred to a PVDF (Polyvinylidene difluoride) membrane using a Thermo Scientific^TM^ iBlot^TM^ 2 dry blotting system. The membranes were then blocked in Odyssey buffer for 4 h. Following incubation with a primary antibody overnight at 4 °C, membranes were washed with PBST (1X phosphate-buffered saline, 0.1% tween^®^ 20) and treated with fluorophore-conjugated secondary antibody for 2 h at room temperature. After washing the membranes, images were scanned using the Odyssey Imaging System and processed, and densitometric analysis was conducted on blots with positive bands using Image Studio Lite software (LI-COR Biosciences, Lincoln, NE, USA).

### 4.7. Quantitative Reverse Transcriptase Polymerase Chain Reaction

LS174T (1 × 10^6^ cells/well) cells were plated in a 12-well plate and incubated overnight. The cells were then treated with either PBS (control) or IL-22 (50 ng/mL) for 12 or 24 h and harvested using TRIzol^TM^ (Invitrogen). In the second set of experiments, LS174T cells or mIECs were pre-treated with ERK1/2 inhibitor (10 μM) for 1 h and them stimulated with IL-22 (50 ng/mL) for 12 h or 4 h, respectively. Pure RNA were collected using the ISOLATE II RNA Mini Kit from Bioline (Alexandria, NSW, Australia). Equal 1 μg of RNA were then used to synthesize corresponding cDNA using a Bioline cDNA synthesis kit. Depending on the targeted genes, the cDNA were diluted up to a 1:10 ratio to perform PCR. Two point five μL of diluted cDNA, 0.75 μL of desired primer (Appendix A), 3.75 μL of SYBR green (SensiFAST^TM^ SYBR^®^ Lo-ROX kit, Bioline), and 0.5 μL of DNase and RNase free water were mixed together and run in a real-time PCR System (Applied Biosystems^®^ ViiA^TM^ 7, Life Technologies Corporation, Camarillo, CA, USA) for 40 cycles. The Ct (cycle threshold) values were then analyzed using a ViiA 7 software (Life Technologies Corporation). The relative quantitations were determined by the ΔΔCt method and normalized to housekeeping gene *β-actin* and expressed as a fold difference to the mean of the relevant control samples. The amplification of the targeted genes were verified from the melting curve obtained from the experiment. The fold-changes were analyzed and plotted using GraphPad Prism software (version 6; GraphPad Software, San Diego, CA, USA).

### 4.8. Transcriptomic Analyses Through RNA Sequencing

To identify cell proliferation and migration-associated transcriptional changes in normal colon epithelium, we performed next-generation-sequencing in mIECs. Following 4 h treatments with either PBS or recombinant mouse IL-22 (100 ng/mL), cells were harvested, and high-quality mRNA were extracted. To confirm activation of the cells, a parallel experiment was conducted to check well-known STAT3 phosphorylation by IL-22. Next generation single-end sequencing was performed by the Australian Genome Research Facility (AGRF) using the HiSeq 2500 machine (Illumina, San Diego, CA, USA) with a maximal read length of 100 bp. The differential gene expression was analyzed through edgeR (version 3.22.3) using R (version 3.5.0; the R Foundation, Vienna, Austria). Raw gene counts per million were used for transcriptional comparison. Gene ontology analyses were done using David, a free online tool to identify and validate (using qRT-PCR) the potential targets associated with cell proliferation and migration.

### 4.9. Statistical Analysis

Data are presented as mean ± SEM. The Statistical Package for Social Science (SPSS22; Chicago, IL, USA) and GraphPad Prism 7 (La Jolla, CA, USA) software programs were employed for the analyses and plotting of the data. One-way ANOVA followed by Dunnett’s post hoc test or two-way ANOVA followed by Bonferroni’s post hoc test were performed wherever applicable to determine statistical differences as indicated in the figure legends.

## Figures and Tables

**Figure 1 ijms-20-03468-f001:**
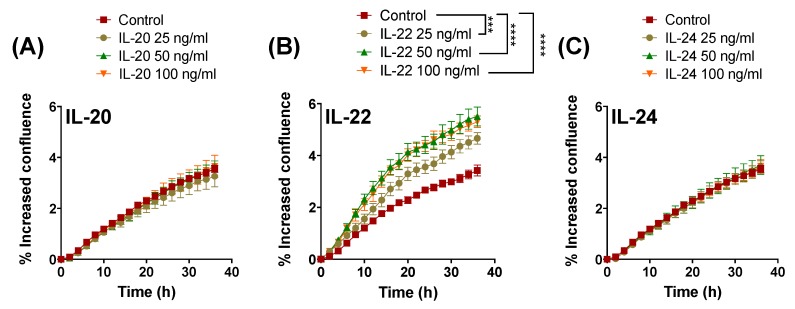
Effects of interleukin (IL)-20 subfamily of cytokines on intestinal LS174T cell proliferation. Cells were treated with 25, 50, or 100 ng/mL of (**A**) IL-20 (**B**) IL-22 or (**C**) IL-24. Cell proliferation was assessed in a live-cell imaging system for 36 h under serum free conditions. Data are presented as mean ± SEM of percentage of increased confluency in each time point of cell proliferation assay. (*n* = 6, two independent experiments). *** *p* < 0.001; **** *p* < 0.0001 compared with control (two-way ANOVA followed by Bonferroni’s post-hoc test).

**Figure 2 ijms-20-03468-f002:**
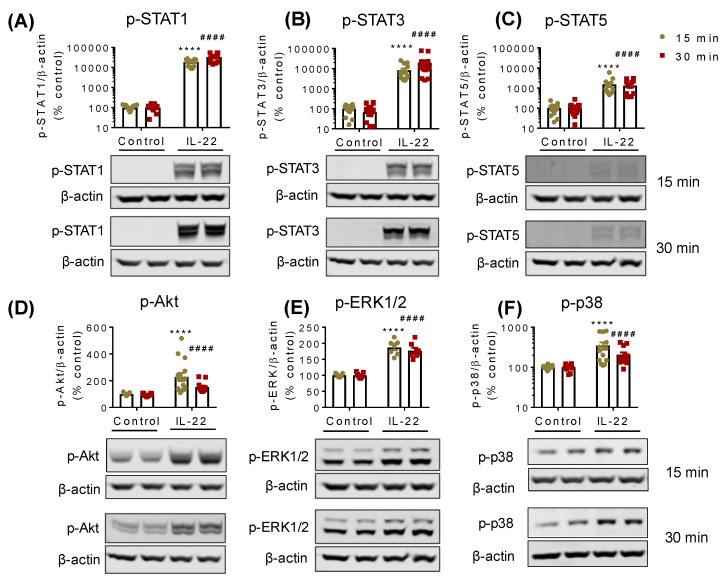
IL-22-activated signalling pathways in intestinal epithelial LS174T cells. Following IL-22 treatment for 15 or 30 min cells were lysed and Western blotting was performed to detect IL-22-induced phosphorylations of (**A**) STAT1, (**B**) STAT3, (**C**) STAT5, (**D**) Akt, (**E**) ERK1/2, and (**F**) p38 in LS174T cells. Data are presented as mean ± SEM with individual cultures (*n* = 13 from four independent experiments). **** *p* < 0.0001 compared with 15 min control and #### *p* < 0.0001 compared with 30 min control (nonparametric Man-Whitney *t*-test).

**Figure 3 ijms-20-03468-f003:**
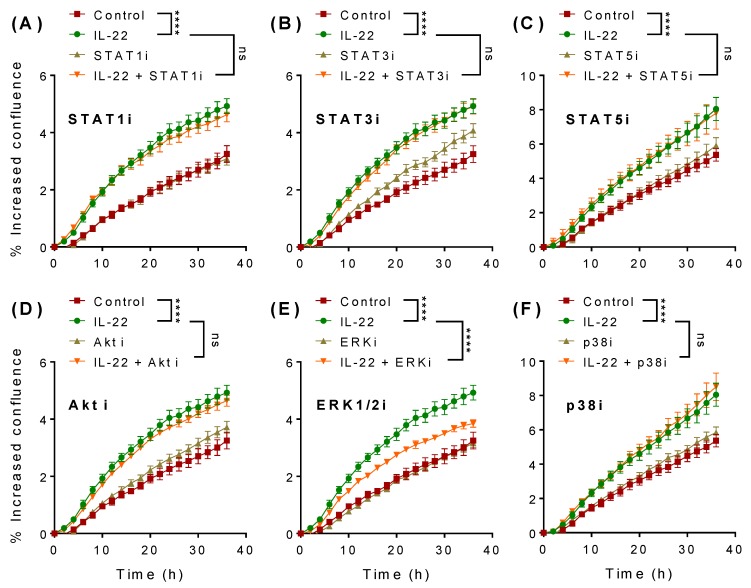
Signaling pathways involved in IL-22-mediated intestinal LS174T cell proliferation. Cells were pre-treated with inhibitors against (**A**) STAT1 (50 μM), (**B**) STAT3 (50 μM), (**C**) STAT5 (50 μM), (**D**) Akt (10 μM), (**E**) p38 (10 μM), or (**F**) ERK (10 μM) alone for 1 h and then observed in the absence or presence of IL-22 (50 ng/mL) for 36 h. Data are presented as mean ± SEM of percentage of increased confluency at each time point of the cell proliferation assay (*n* = 6, two independent experiments). **** *p* < 0.0001 compared with control or IL-22 as indicated (two-way ANOVA followed by Bonferroni’s post-hoc test).

**Figure 4 ijms-20-03468-f004:**
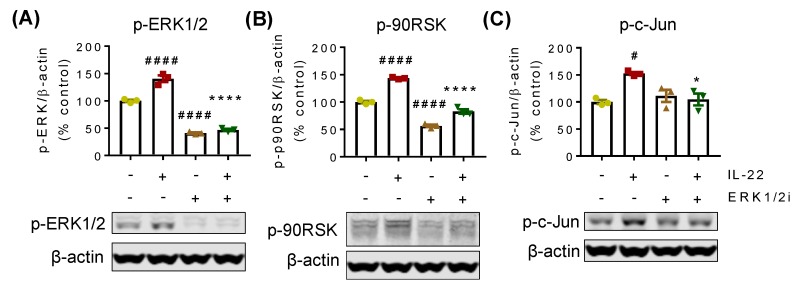
Activation of p90RSK and c-Jun by ERK1/2. LS174T cells were treated with 50 ng/mL of IL-22 for 30 min following 1 h pre-treatment with ERK1/2 inhibitor (10 μM). Cells were lysed and Western blotting was performed to detect phosphorylated (**A**) ERK1/2, (**B**) p90RSK, and (**C**) c-Jun. Data are presented as mean ± SEM with individual cultures (*n* = 3). # *p* < 0.05; #### *p* < 0.0001 compared with control and * *p* < 0.05; **** *p* < 0.0001 compared with IL-22 (one-way ANOVA followed by Dunnett’s post-hoc test).

**Figure 5 ijms-20-03468-f005:**
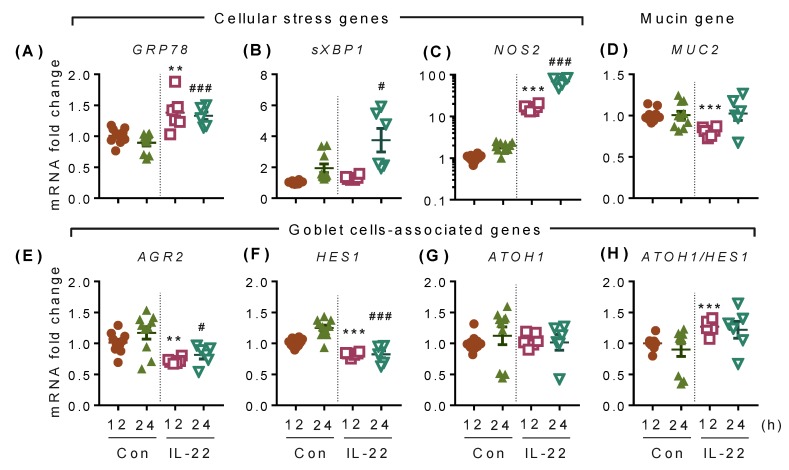
IL-22-mediated transcriptional changes in intestinal LS174T cells. Cells were treated with IL-22 (50 ng/mL) for 12 and 24 h and harvested to collect RNA. Then, qRT-PCR was performed to check the expression of ER stress markers (**A**) GRP78 and (**B**) sXBP1, (**C**) oxidative stress marker NOS2, (**D**) mucin MUC2, goblet cell-associated factors (**E**) AGR2, (**F**) HES1, (**G**) ATOH1, and (**h**) the ATOH1/HES1 ratio. Data are presented as mean ± SEM with individual cultures (*n* = 6–9 from 2–3 independent experiments). ** *p* < 0.01; *** *p* < 0.001 compared with 12 h control and ### *p* < 0.001 compared with 24 h control (nonparametric Man-Whitney *t*-test).

**Figure 6 ijms-20-03468-f006:**
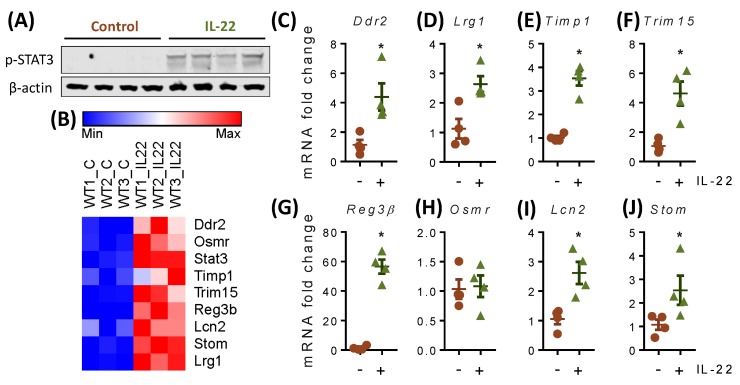
IL-22-mediated transcriptional changes in primary epithelial cells. Murine intestinal epithelial cells (mIECs) were treated with 100 ng/mL of mIL-22 for 4 h and (**A**) activation of STAT3 was assessed using Western blotting. RNA-Seq was conducted and (**B**) gene ontology analyses showed differential expression of antimicrobial peptides, glucose transporter, and proliferation-migration associated genes with IL-22. qRT-PCR was performed to validate the expression of (**C**) Ddr2, (**D**) Lrg1, (**E**) Timp1, (**F**) Trim15, (**G**) Reg3β, (**H**) Osmr, (**I**) Lcn2, and (**J**) glucose transporter Stom. Data are presented as mean ± SEM with individual primary cultures from different mice (*n* = 4). * *p* < 0.05 (nonparametric Man-Whitney *t*-test).

**Figure 7 ijms-20-03468-f007:**
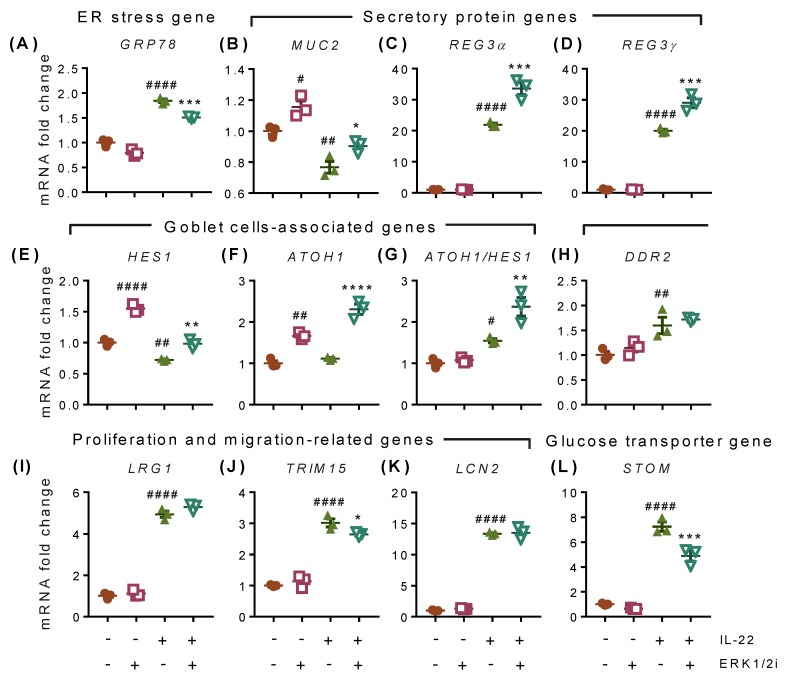
IL-22-mediated transcriptional changes in intestinal LS174T cells. Cells were pre-treated with 10 μM of ERK1/2 inhibitor for 1 h before treatment with IL-22 (50 ng/mL) for 12 h. mRNA expression of (**A**) ER stress marker GRP78, (**B**) mucin gene MUC2, antimicrobial peptides (**C**) REG3α and (**D**) REG3γ, (**E**) goblet cells-associated factor HES1, (**F**) cell differentiation marker ATOH1, (**G**) ATOH1 and HES1 ratio, cell proliferation-associated genes (**H**) DDR2, (**I**) LRG1, (**J**) TRIM15, and (**K**) LCN2, and (**L**) glucose transporter STOM. Data are presented as mean ± SEM with individual cultures (*n* = 3 from one experiment). # *p* < 0.05; ## *p* < 0.01; #### *p* < 0.0001 compared with control and * *p* < 0.05; ** *p* < 0.01; *** *p* < 0.001 compared with IL-22 (one-way ANOVA followed by Dunnett’s post hoc test).

**Figure 8 ijms-20-03468-f008:**
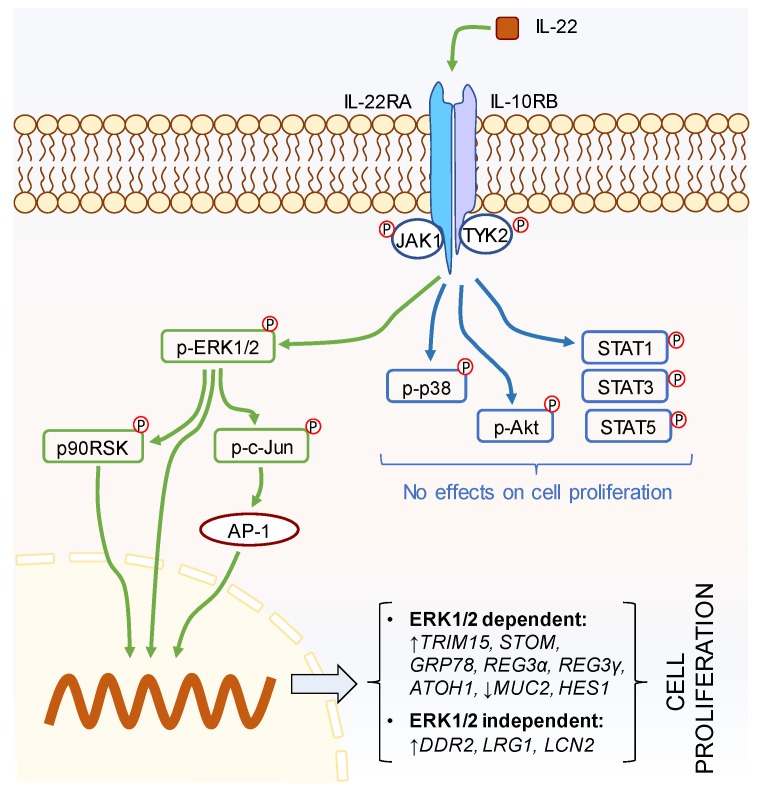
IL-22-induced modulation of intestinal epithelial cell proliferation is dominated by the ERK1/2 pathway. Green arrows indicate possible mechanism(s) of IL-22, where blue arrows show no effects on cell proliferation.

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
