# Peer review of "Interleukin (IL)-22 from IL-20 Subfamily of Cytokines Induces Colonic Epithelial Cell Proliferation Predominantly through ERK1/2 Pathway"

_ijms, 2019, doi:10.3390/ijms20143468_

Round 1
Reviewer 1 Report
Review of manuscript: Interleukin (IL)-22 from IL-20 subfamily of cytokines induce colonic epithelial cell proliferation predominantly through ERK1/2 pathway
The current study examined the effects of members of the IL-20 cytokine family that are known to be overexpressed in the colon of patients with ulcerative colitis. The effect of IL-20, IL-22, and IL-24 on the proliferation and expansion of epithelial cells were determined using the colonic and goblet-like cell line LS174T and only IL-22 was found to enhance the proliferation of these cells. A number of pathways were shown to be activated by IL-22 and ERK1/2 was found to be the predominant pathway that regulated cell proliferation. Using immunoblot analyses and transcriptional studies of target genes and gene ontology analyses, the authors identify transcriptional mediators that promote cell proliferation and migration as well as genes involved in the regulation of cell differentiation that were responsive to IL-22.
It should be noted that most of the study was performed in vitro using a single cell line, while RNA sequencing studies were carried out in primary mouse epithelial cells to identify genes involved in cell proliferation and migration through gene ontology studies, presumably to show that the observations extend in vivo. These studies further suggest the involvement of IL-22 IN ER stress response and cellular stress in proliferating cells through LCN2.
Reviewer comments:
In general, the manuscript was well written and the conclusions were consistent with the experimental results and observations. Overall, there were a number of genes that were studied and correlated with various cellular processes including proliferation, cell expansion, migration and stress responses that were proposed to be affected by IL-22. While the discussion proposes some interesting mechanisms underlying these processes based on gene expression studies, in light of the large number of genes and processes that have been analyzed, it would be useful if the authors include a figure (ie, Figure 8) with a schematic diagram illustrating the effect of IL-22 on these processes that delineates the transcription factors and genes that they propose to be mediators of these processes.
Reviewer 2 Report
Moniruzzaman et al. demonstrate in this paper that among the IL-20 cytokine subfamily, only IL-22 but not IL-20 or IL-24 could increase cell proliferation rate of a human colon adenocarcinoma cell line, LS174T. ERK1/2 might play a role in this effect because growth rate of LS174T cells was decreased in the presence of ERK1/2 inhibitors. IL-22 stimulation induced change in expression of various genes in LS174T cells as well as primary mouse intestinal epithelial cells (IELs). The results are clear; however, I have some issues to be clarified so that the conclusion of this paper would be more convincing and reliable.
1. In Figure 1, the authors showed only IL-22 could promote cell growth of LS174T. I wonder why IL-20 or IL-24 did not stimulate LS174T cell proliferation. To clarify whether LS174T cells express receptors for IL-20 and IL-24, the authors should examine expression of IL-20RA, IL-20RB, IL-22RA and IL-10RB. The authors may also examine expression of these receptor subunits in mouse IELs. Based on the results, the authors should add their interpretation to the Result and Discussion sections.
2. It is reasonable to apply the evidence obtained with cell lines to primary cells. But in this case, one need to verify whether characteristics of the cell lines can apply to primary cells. For this study, the authors clarified IL-22 is unique in terms of ability to accelerate proliferation rate of LS174T cells. Oddly, the authors did not verify this unique characteristic in primary IELs. Similarly, although the authors examined the effect of ERK1/2 inhibitors in LS174T cells but not in primary mouse IELs. The authors should perform these experiments and discuss similarity and/or difference between LS174T cells and mouse IELs depending on results of the experiments.
Round 2
Reviewer 2 Report
As I pointed in my previous review, I would like the authors to discuss why only IL-22 but not IL-20 or IL-24 can exert the function in LS174T cells and mIECs when the receptors for these 3 cytokines are expressed.
Author Response
We thank the reviewer for their comments. Although belonging to the same sub-family, previous literature also corroborates our findings of the potent cell proliferative roles of IL-22 as compared with IL-20 and IL-24. We have now speculated as to what the reason might be for the lack of an effect of IL-20 and IL-24 on cell proliferation (lines 238-245), and hope that this is clearer for the readers.